# Whole Spine and Sacroiliac Joint MRI of Patients With and Without Metabolic Syndrome: A Preliminary Study

**DOI:** 10.3390/diagnostics16010108

**Published:** 2025-12-29

**Authors:** Amir Bieber, Shay Brikman, Mohamad Nujeidat, Irina Novofasovsky, Reuven Mader, Iris Eshed

**Affiliations:** 1Rappaport Faculty of Medicine, Technion Israel Institute of Technology, Haifa 3109601, Israel; shay.brikman@technion.ac.il; 2Rheumatic Diseases Unit, Emek Medical Center, Afula 18101, Israel; irina_no@clalit.org.il (I.N.); einatmader@gmail.com (R.M.); 3Department of Internal Medicine C, Emek Medical Center, Afula 1834111, Israel; nujeidatmohamad@gmail.com; 4Department of Diagnostic Imaging, Sheba Medical Center, Ramat Gan 5262160, Israel; iriseshed@gmail.com; 5The Gray Faculty of Medical and Health Sciences, Tel Aviv University, Tel Aviv 69978, Israel

**Keywords:** DISH, MRI, metabolic syndrome

## Abstract

**Background:** MRIs of spine Diffuse Idiopathic Skeletal Hyperostosis (DISH) demonstrate inflammation-related lesions such as bone marrow edema (BME) and fatty lesions in vertebral corners. DISH is known to be associated with metabolic syndrome (MeS). Spinal MRI lesions in patients with metabolic syndrome (MeS) alone were not described. **Aim:** To characterize spine and sacroiliac MRI lesions among patients with MeS. **Methods:** This study is a small preliminary cross-sectional case–control study. Study groups were defined as patients with and without MeS. All patients were between 40 and 50 years of age and did not have DISH by chest radiography or CT. All patients underwent whole-spine and sacroiliac joint MRI. MRI was scored by a radiologist blinded to the patient’s clinical data for the presence of BME, fat lesion, sclerosis, ankylosis, erosions, enthesitis, and capsulitis. Groups were compared for the prevalence of each lesion, and scores were calculated for inflammatory and structural scores. Clinical data regarding MeS was also collected. **Results:** Twenty-four patients were included (twelve male; mean age: 45 years at imaging); fifteen with and nine without MeS. Patients in the MeS study group had significantly more MRI lesions, reflected as a higher total sum lesion score (*p* = 0.013), a higher fat lesion score (*p* = 0.013), and a higher total structural lesion score (*p* = 0.014). Also, each of the MeS components was significantly associated with higher MRI scores. **Conclusions:** Significantly higher spinal and sacroiliac scores of both inflammatory- and structural-related lesions were present on MRI of MeS patients compared to patients without MeS.

## 1. Introduction

Diffuse Idiopathic Skeletal Hyperostosis (DISH) is a hyperostotic syndrome first described in the mid-20th century [1]. It is characterized by gross calcification and ossification, mostly of the thoracic spine [2]. The association between DISH and metabolic syndrome (MeS) is well known [3]. Several factors associated with MeS are specifically associated with DISH, mainly obesity, type 2 diabetes mellitus (DM), and cardiovascular atherosclerotic morbidity [4,5,6,7,8]. While DISH represents a relatively end-stage of axial ossification, pre-DISH states have been described, mostly using chest radiography or computed tomography (CT) [9,10]. However, there are currently no studies that specifically address the pre-DISH state and MeS on MRI.

Both DISH and axial spondyloarthritis (axSpA) are enthesis-related bone-forming conditions that in their end stage may lead to spinal ankylosis [11]. Nevertheless, their pathophysiology is thought to differ substantially: DISH is dominated by structural new bone formation, whereas axSpA is characterized by inflammation-driven lesions that may precede structural change [12]. MRI lesions can therefore be conceptually grouped into inflammatory (active) findings—such as bone marrow edema (BME), enthesitis, and capsulitis—and structural findings—such as fat lesions, sclerosis, erosions, and ankylosis.

In established DISH, prior MRI studies have mainly described structural corner-based changes (particularly fat lesions and sclerosis), with the variable presence of BME, and overall patterns that can differ from axSpA/AS, particularly in the distribution and extent of ankylosis [13]. However, these reports address established DISH and do not clarify whether similar MRI abnormalities can be detected earlier in individuals with MeS in the absence of radiographic/CT evidence of DISH.

The spinal MRI of patients with MeS has been explored in limited settings, including postmenopausal women with DM, where lumbar MRI spectroscopy suggested altered marrow lipid composition and higher marrow density consistent with the skeletal effects of DM [14,15]. In addition, ultrasound studies suggest that MeS may be associated with a pro-inflammatory entheseal phenotype and a proportion of these patients also fulfill DISH criteria [16].

The presence and pattern of spinal MRI lesions in patients with MeS without imaging evidence of DISH (i.e., “pre-DISH” candidates) have not yet been studied. This study aims to describe patterns of MRI lesions among relatively young MeS patients with no imaging evidence of DISH. We hypothesized that the MeS group would have higher MRI lesion scores compared to the non-MeS group.

## 2. Materials and Methods

This is a cross-sectional study comparing the frequency of MRI lesions on the spine and sacroiliac MRI of two small groups of patients with and without MeS. The presence of MeS was defined according to the National Cholesterol Education Program, Adult Treatment Panel III (NCEP 3) [17].

### 2.1. Patient Population

All patients were recruited from the Rheumatology unit at Emek Medical Center (Afula, Israel) between 2018 and 2022 on a voluntary basis. Most patients were evaluated for non-inflammatory back pain; others were recruited following publication for volunteers for the trial and were otherwise asymptomatic. Inclusion criteria for all patients aged between 40 and 50 years, did not fulfill the classification criteria for DISH by chest radiography, or had CT performed within a maximum of one year before recruitment. We focused on this relatively narrow age range because DISH is presumed to be less common at these ages, whereas MeS is still prevalent, thus allowing MeS to be evaluated as a potential pre-DISH state. Excluded were patients with inflammatory back pain, spondyloarthropathy, uveitis, psoriasis, rheumatoid arthritis, inflammatory bowel disease, HLA-B27 positivity, and family history of spondyloarthropathies. The patients were referred to this clinic for primary evaluation of musculoskeletal complaints or were recruited voluntarily without specific medical complaints of problems.

### 2.2. MRI Scans and Evaluation

All patients underwent spine and SIJ MRI using either a 3.0T scanner (Inginia, Philips healthcare, Amsterdam, The Netherlands) utilizing a phased-array 16-element coil, including the following sequences: for the entire spine, sagittal T1-W and T2-W, and for the SIJ, a semi-coronal T1-W and T2-W with fat saturation.

All MRI scans were evaluated by an experienced MSK radiologist (IE, 21 years of dedicated MSK experience) who was blinded to patients’ clinical data.

Findings in the spine and SIJ were scored binarically for the presence or absence of structural (erosion, fat lesion, sclerosis, ankylosis) and inflammatory (BME, enthesitis, capsulitis) lesions using a slight modification of the Berlin scoring method [18]. Because the Berlin scoring system was originally developed to quantify spinal and sacroiliac joint inflammation in axial spondyloarthritis, its grading relies on the presence of relatively large and frequent inflammatory lesions. In contrast, patients with DISH typically demonstrate much smaller, more focal, and less frequent lesions, making the full 0–3 Berlin scale disproportionate to this population. To maintain the same anatomical topography and standardized assessment framework while ensuring sensitivity to the subtle lesions characteristic of DISH, we adapted the scoring as follows: sacroiliac joints were scored on a binary 0–1 scale (absent/present) and vertebral corner lesions in the spine on a 0–2 scale reflecting absent, mild, or definite focal abnormalities. This modification preserves consistency with established axSpA scoring methodology while improving its applicability and discriminatory capacity in DISH.

### 2.3. Scoring Description

SIJ Scoring (total: 48 points): SIJs were scored binarically for the presence (1) or absence (0) of a lesion.

•BME, fat lesion, erosion, and sclerosis:

◦One point each for the following regions (eight regions in total): anterior and posterior, lower and upper parts, for both ilium and sacrum.◦Total for these changes: 8 points each (32 points in total).

•Ankylosis:

◦One point each for the same eight regions (anterior/posterior, lower/upper, ilium/sacrum) plus two points for anterior bridges.◦Total for ankylosis: 10 points.

•Enthesitis and Capsulitis:

◦Two points for each side (right and left).◦Total for enthesitis and capsulitis: 4 points.

Spine Scoring (Total: 92 points)

•For each vertebral unit (two adjacent vertebrae around the disk space), score 0–2 points for BME, fat lesion, erosion, posterior elements inflammation, and syndesmophytes.•Twenty-three vertebral units, scored for vertebral body and posterior elements.

A structural lesion score (sum of erosions, sclerosis, fat lesion, syndesmophytes, and ankylosis), and active lesion score (BME and enthesitis and capsulitis), and a total sum score (structural and active) were calculated for each patient’s MRI examination.

Clinical data regarding MeS was also collected included: MeS component, BMI, abdominal circumference, serum lipid profile, serum insulin levels at time of recruitment, serum uric acid, HbA1C% status, and the presence of comorbidities at the time of recruitment (DM, Hypertension (HTN), Dyslipidemia, Chronic Obstructive Pulmonary Disease, Ischemic Heart Disease, Gout, and chronic kidney disease). Also, chronic medication intake was collected, as well as routine chemistry and blood count values. All clinical data was collected from electronic medical records with data available in a time frame of 6 months before and after the study MRI.

Statistics: Demographic and clinical characteristics of the two groups were compared by Fisher’s exact test for categorical variables and by *t*-test or Mann–Whitney test in the case of non-normally distributed data for continuous variables. Spearman correlations between age, MeS components, and laboratory results were performed. To isolate independent data that influences the results, *p*-values were used. Statistical significance was determined by *p* < 0.05. Two-sided analysis was used as the direction of the effect was not pre-determined.

### 2.4. Ethics

The study was performed according to the International Conference on Harmonization (ICH), the Harmonized Tripartite Guideline for Good Clinical Practice (GCP), and according to the procedures of the Ministry of Health for conducting medical research.

This study was approved by the Emek Medical Center Helsinki Committee for Research (approval no. 17-013). Written informed consent was approved by the committee and obtained from all participants.

## 3. Results

Twenty-four patients were included (thirteen males; mean 45.5 years at imaging, range 40–49 years), fifteen in the MeS group and nine in the non-MeS group. The cohorts’ demographics, medical history, medications and blood chemistry data are presented in Table 1 and Table 2. The mean age was 44.9 ± 3.1, without a statistically significant difference. Among the MeS group, 9 out of 15 were males compared to controls with 3 out of 9, without statistically significant difference. Patients with MeS had a statistically significantly higher rate of all MeS components (i.e., higher triglycerides and lower HDL, higher frequency of DM and IFG, larger waist circumference, and significantly higher BMI (34.9 ± 6.3 vs. 22.7 ± 1.6). There were no other statistically significant differences in medical history or any significant differences in medication uptake between the two groups. Patients with MeS also had significantly higher median insulin, uric acid, AST, ALT, ESR and CRP levels.

### MRI Scores

Table 3 presents the MRI scores divided according to spinal location and SIJ. Examples of the different MRI findings are presented in Figure 1, Figure 2 and Figure 3. Cervical and thoracic spine scores of patients with MeS were higher, though without statistical significance, for total sum scores as well as total structural scores. Lumbar spine total sum score was higher among MeS vs. non-MeS, with *p* = 0.034. All other MRI scores of the lumbar spine were higher, yet not statistically significant. Of note, nearly significantly higher fat lesion and ankylosis scores were registered in the lumbar spine. It should be mentioned that for two-sided *p*-value analysis, the cervical total sum score, cervical total structural score, thoracic total sum, structural, and fat lesion scores all had a tendency toward significance (*p* between 0.05 and 0.07). There was no statistically significant difference in total sum score, structural, active or specific lesion scores of the SIJs.

Table 4 presents the MRI scores for the entire spine and SIJ. There was a statistically significant difference in the MRI total sum score of the entire spine and SIJ between the two groups, with patients in the MeS group having a significantly higher value (7.7 ± 8.0 vs. 0.6 ± 0.8, *p* = 0.013 for the MeS vs. non-MeS group, accordingly). In addition, patients in the MeS group had a significantly higher total fat lesion score (2.47 ± 3.13 vs. 0.22 ± 0.67, *p* = 0.013, for the MeS vs. non-MeS group, accordingly) and total structural lesion score (6.27 ± 7.26 vs. 0.33 ± 0.71, *p* = 0.014, for the MeS vs. non-MeS group, accordingly).

HTN was associated with MRI total sum score (*p* = 0.021), MRI total ankylosis score (*p* = 0.009), and MRI total structural score (*p* = 0.044). DM was associated with MRI total enthesis score (*p* = 0.008) and MRI total inflammation score (*p* = 0.044).

Large waist was associated with MRI total sum score (*p* = 0.028), MRI total fat lesion score (*p* = 0.023), MRI total ankylosis score (*p* = 0.023), and MRI total structural score (*p* = 0.028). Chronic kidney disease was associated with MRI total structural score (*p* = 0.044). MRI total ankylosis score was negatively correlated with age. MRI total fat lesion score was positively correlated with CRP and uric acid. MRI scores were not associated with sex, smoking, IFG, OSA, or back pain.

## 4. Discussion

In this descriptive, relatively small cross-sectional study, we assessed spine and SIJ MRIs of subjects with and without MeS. Our main observation was that the total sum MRI score, total fat lesion score, and structural MRI score were higher in subjects with MeS compared to those without. The summation method was used to emphasize the whole-spine effect of our cohort. Although this study involves a relatively small cohort, it provides important insight into a topic that is not yet fully understood, which is MRI lesions of the axial skeleton associated with MeS and the relationship to the pathogenesis of DISH. We hypothesized that even in young patients with MeS but without DISH, MRI findings that might be indicative of a pre-DISH state would be present, potentially suggesting an early metabolic-driven cascade of bone formation. It should be noted that not all MeS are also defined as pre-DISH states, and not all pre-DISH will eventually progress to DISH fulfilling Resnick’s criteria. Hence, the results of this study focus mostly on the association between MeS and spinal MRI changes that might precede pre-DISH and DISH.

Indeed, we found a significantly higher total score and structural score among MeS compared to the non-MeS group. When focusing on the spine location, the thoracic spine among MeS contributed the most for fatty lesions, while all other spinal portions contributed relatively equally to both BME and ankylosis (Table 3). Studies focusing on the MRI of patients with DISH describe the presence of structural and inflammatory-associated lesions like those found in axSpA, albeit occurring at a lower frequency [13,19,20]. In these studies, most of the spinal lesions were structural, just like those seen in our study on a cohort of speculated pre-DISH state of MeS patients. The combination of these and our results supports the hypothesis that MeS plays a role in the pathogenesis of bone formation and structural alterations in the axial skeleton in DISH. Yet previous studies regarding MRI findings of spines among patients with MeS did not report systematic scoring in terms of inflammatory or structural changes, so there is no reference for comparison of our results [14,21,22].

As first described by Resnick et al., DISH predominantly affects the thoracic spine but may also involve the cervical and lumbar spine [1], while the thoracic spine is less affected by degenerative spondylosis compared to the cervical and lumbar segments [23]. This distribution was not observed in our study, where structural lesions were more prevalent in the thoracic spine compared to the cervical and lumbar segments. This finding suggests that MeS might contribute to pre-DISH pathogenesis in a less frequently degenerated thoracic spine. It should be noted that Resnick et al.’s works predominantly focused on a well-established DISH population, while we were looking at a small cohort of patients with MeS vs. normal controls, all between 40 and 50 years of age.

BME lesions were much less prevalent compared to structural ones, while structural lesions were similarly prevalent in both study groups across all spinal segments (Table 3 and Table 4). Indeed, this was also reported by Ziade et al., highlighting less inflammation in the spine of patients with DISH compared to structural lesions and compared to the main role of BME in axSpA, potentially indicating a different etiopathogenesis [20]. A recent study by Keifer et al., examined the associations of radiographic (hence structural) lesions with MeS components among DISH vs. non-DISH and found a strong correlation between the degree of obesity and structural changes. This study further supports the connection between structural changes seen radiographically and obesity as part of the MeS [24].

In the current study, there was no difference in the lesions’ score in the SIJ between MeS and non-MeS patients. This is contrary to studies focusing on patients with DISH in which BME was found in the majority of the SIJ [20]. A study by Latourte et al. exploring the MRI of the SIJ among patients with DISH revealed inflammatory lesions, including BME at 18% of patients, and capsulitis and enthesitis in 2.6% and 5.5%, accordingly, while structural lesions such as fatty lesions were demonstrated in 33% of patients and ankylosis in 20% [25]. The difference from our study likely stems from the distinct cohorts; a pre-DISH with MeS vs. DISH subjects potentially reflects variations in MRI lesions at different stages of DISH evolution.

There was a significant correlation between HTN, DM, waist circumference, and MRI scores, mainly the total sum score. An association of BMI and higher fatty lesion scores was also found. Associations between MeS and DISH are well described, while the specific contribution of each component of MeS might be different. In previous reports, we found an extremely high prevalence of DISH among patients with obesity hypoventilation syndrome (more than 90%) and among young patients with morbid obesity [5,6]. The specific contribution of weight to MRI lesions might be even higher than other factors of MeS, such as DM, HTN, or lipid level, while this topic, specifically at pre-DISH states, remains to be explored. Evidence from systematic evaluations of spinal and SIJ MRI changes in patients with MeS remains limited. Badr et al. used MRI with spectroscopy to assess bone marrow adiposity in postmenopausal women with type 2 DM, demonstrating lower levels of unsaturated fat in the lumbar spine in this population [14]. In addition, Woods et al. reported a negative correlation between unsaturated fat levels in bone marrow and fracture risk. However, neither study assessed concomitant morphological spinal changes or soft-tissue alterations related to adiposity [15]. Moreover, no additional studies evaluating the association between MeS and MRI-detected spinal changes were identified, positioning our work as preliminary and underscoring the need for confirmation in larger, well-characterized cohorts. The potential emerging pharmacological treatments for metabolic syndrome, especially for obesity and DM with glucagon-like peptide-1 receptor agonists (GLP1A), might promise a secure way for prevention of bone formation in this context, a hypothesis that should be further explored [26].

Our study limitations include a relatively small group of patients, with some major differences between groups, especially BMI. Because our groups were small and we analyzed multiple MRI lesion types across several regions, the MRI changes reported here may not be directly attributable to MeS itself. Instead, they could reflect the influence of individual MeS components and the differences in these components between groups. Accordingly, these findings are susceptible to both type I and type II errors and should be interpreted with caution. Yet, our study is a preliminary work about the associations between MeS and MRI spinal lesions, and despite its small number, do point to some specific findings. The systematic analysis of the whole spine resulted in a large-scale comparison that yielded statistically meaningful changes. Another limitation concerns the different aspects of MeS itself within the MeS group. For example, only a minority of our patients had DM, which may have also biased our results regarding its treatment. Further studies need to be performed within different MeS components to better understand each of these musculoskeletal findings with MRI. Another limitation is that participants were recruited through a rheumatic disease unit. Although enrollment was voluntary and included individuals both with and without back pain, this recruitment strategy may have introduced selection bias. To reduce this risk, we systematically evaluated for inflammatory back pain and excluded participants who fulfilled established criteria. Finally, as this was a cross-sectional analysis at one point in time, no longitudinal data was available to address the important issue of ossification progression and its biomarkers. Also, as progression to DISH takes years, which study could not address, since we allowed no more than 12 months of CT or XR before inclusion.

## 5. Conclusions

Significantly higher MRI scores related to spine but not SIJ lesions were present on MRI of MeS patients compared to patients without MeS, potentially suggesting a pathogenetic linkage between MeS and spinal bone formation. This is the first MRI study among MeS patients, pointing to a presumed earlier stage of DISH before the complete ossification process occurs. Further studies are needed to elucidate the pathogenesis associated with such structural lesions.

## Figures and Tables

**Figure 1 diagnostics-16-00108-f001:**
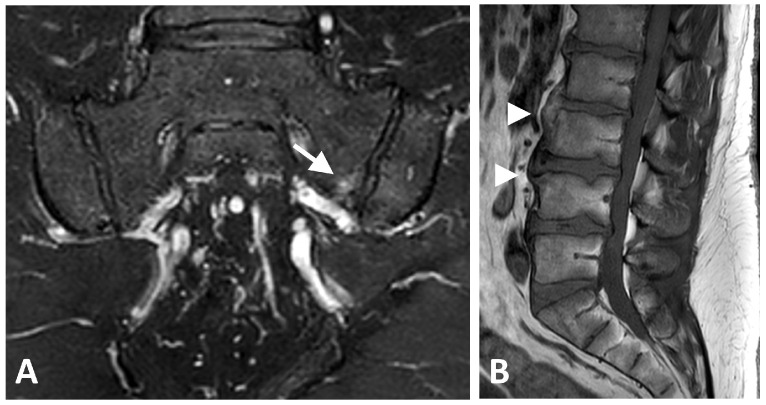
MRI images of a 42-year-old female with metabolic syndrome. (**A**) Semicoronal T2-weighted fat-saturated image of the sacroiliac joints demonstrating mild bone marrow edema (score 1) on the left sacral side of the joint (arrow). (**B**) Sagittal T1-weighted image of the lumbar spine showing anterior flowing osteophytes at the L2–L3 and L3–L4 intervertebral levels (arrowheads). No additional active or structural lesions were identified in the sacroiliac joints or the spine.

**Figure 2 diagnostics-16-00108-f002:**
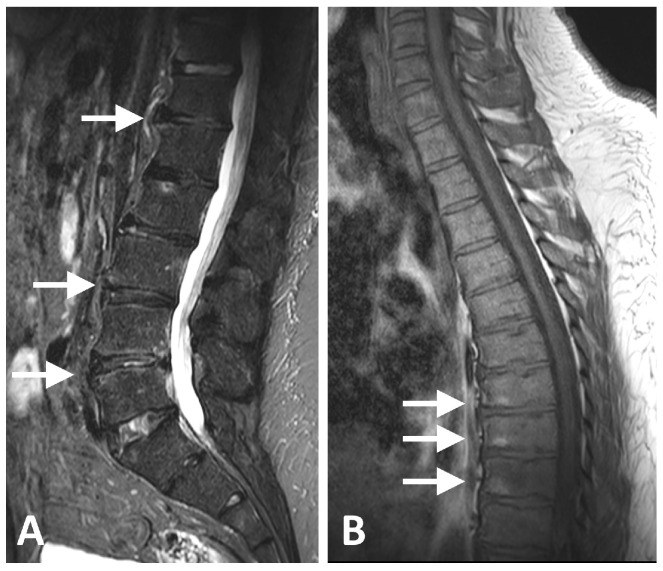
MRI images of a 49-year-old male with metabolic syndrome. (**A**) Sagittal T2-weighted fat-suppressed image of the lumbar spine and (**B**) T1-weighted image of the thoracic spine demonstrating small anterior corner bone marrow edema lesions (score 1) at the D12-L1, L3–L4 and L4–L5 intervertebral levels (arrows in (**A**)) and anterior fat lesions (score 1) at the D7–D8, D8–D9, and D9–D10 intervertebral levels (arrows in (**B**)). No active or structural lesions were identified in the sacroiliac joints.

**Figure 3 diagnostics-16-00108-f003:**
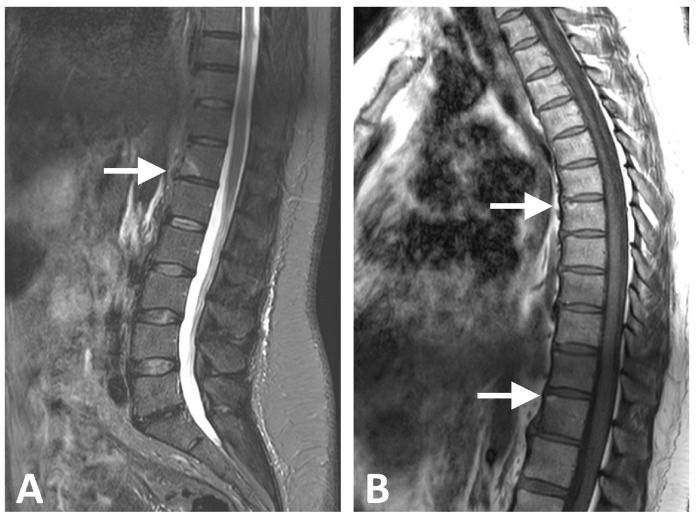
MRI images of a 49-year-old female with metabolic syndrome: (**A**) Semicoronal T2-weighted fat-saturated image demonstrating a large, anterior corner bone marrow edema lesion (score 2) at the D12–L1 intervertebral level (arrow). (**B**) Sagittal T1-weighted image of the lumbar spine showing small anterior fat lesions at the D5–D6 and D10–D11 intervertebral levels (arrows). No active or structural lesions were identified in the sacroiliac joints.

**Table 1 diagnostics-16-00108-t001:** Patient characteristics with and without metabolic syndrome.

Characteristic	All (*N* = 24)	Metabolic Syndrome (*N* = 15)	No Syndrome (*N* = 9)	*p* Value
Age at imaging, years, mean ± SD (range)	44.9 ± 3.1 (40–49)	45.4 ± 3.2 (40–49)	43.7 ± 3.1 (40–49)	0.21
Male, *n* (%)	12 (50.0)	9 (60.0)	3 (33.3)	0.40
IFG, *n* (%)	9 (37.5)	9 (60.0)	0 (0.0)	0.007
Large waist circumference, *n* (%)	14 (58.3)	14 (93.3)	0 (0.0)	<0.001
BMI, mean ± SD (range)	30.3 ± 7.8 (18.7–48.4)	34.9 ± 6.3 (21.8–48.4)	22.7 ± 1.6 (18.7–24.0)	<0.001
Hypertension, *n* (%)	5 (20.8)	5 (33.3)	0 (0.0)	0.12
Triglycerides, mg/dL, mean ± SD (range)	170.6 ± 172.3 (43–806)	223.5 ± 200.2 (58–806)	82.6 ± 35.7 (43–151)	0.003
HDL, mg/dL, mean ± SD (range)	50.2 ± 10.4 (32–73)	45.3 ± 6.5 (32–59)	58.3 ± 10.9 (41–73)	0.007
Type 2 diabetes, *n* (%)	3 (12.5)	3 (20.0)	0 (0.0)	0.27
IHD, *n* (%)	7/23 (30.4)	7/14 (50.0)	0 (0.0)	0.019

IFG—Impaired Fasting Glucose, BMI—Body Mass Index, HDL—High Density Lipoprotein, IHD—ischemic heart disease, Large waist circumference according to Metabolic Syndrome Definitions [17].

**Table 2 diagnostics-16-00108-t002:** Laboratory results related to metabolic syndrome.

Laboratory Parameter	All (*N* = 24)	Metabolic Syndrome (*N* = 15)	No Syndrome (*N* = 9)	Test Statistic	*p* Value
Insulin, µU/mL, mean ± SD (range)	14.45 ± 18.02 (0.1–196)	18.90 ± 24.40 (3.4–196)	6.30 ± 7.43 (0.1–24.0)	2.80	0.004
HbA1c, %, mean ± SD (range)	5.4 ± 0.5 (4.3–8.8)	5.4 ± 0.8 (4.8–8.8)	5.4 ± 0.5 (4.3–5.6)	0.76	0.45
A1C (highest), %, mean ± SD (range)	5.6 ± 0.7 (4.3–14.8)	5.7 ± 0.9 (4.8–14.8)	5.4 ± 0.6 (4.3–5.8)	2.03	0.04
Creatinine, mg/dL, mean ± SD (range)	0.788 ± 0.204 (0.52–1.36)	0.807 ± 0.204 (0.55–1.36)	0.757 ± 0.212 (0.52–1.14)	0.58	0.57
Uric acid, mg/dL, mean ± SD (range)	5.36 ± 1.37 (2.42–7.20)	6.04 ± 1.02 (3.58–7.20)	4.20 ± 1.09 (2.42–5.49)	3.72	0.002
AST, U/L, mean ± SD (range)	22.0 ± 5.8 (13–35)	24.0 ± 6.1 (14–35)	18.8 ± 3.2 (13–23)	2.36	0.028
ALT, U/L, mean ± SD (range)	23.4 ± 10.3 (9–50)	27.6 ± 10.1 (10–50)	16.3 ± 6.2 (9–27)	3.01	0.006
LDL, mg/dL, mean ± SD (range)	109.6 ± 25.0 (70–170)	112.4 ± 26.1 (70–170)	100.4 ± 17.8 (85–141)	1.18	0.25
Hemoglobin, g/dL, mean ± SD (range)	13.5 ± 1.2 (11.3–16.6)	13.7 ± 1.3 (11.3–16.6)	13.3 ± 1.0 (12.0–15.1)	0.82	0.42

Abbreviations: AST—Aspartate Aminotransferase; ALT—Alanine Aminotransferase; LDL—Low-Density Lipoprotein.

**Table 3 diagnostics-16-00108-t003:** MRI scores (median; range) by location and by group. P-s are 2-tailed from the Mann–Whitney test.

	SIJ				Cervical				Thorax				Lumbar			
MRI	All	MeS	No MeS	*p*	All	MeS	No MeS	*p*	All	MeS	No MeS	*p*	All	MeS	No MeS	*p*
Sum of all scores	0.12 ± 0.33 (0; 0–1)	0.19 ± 0.39 (0; 0–1)	0.00 (0; 0–0)	0.67	1.38 ± 3.24 (0; 0–12)	2.00 ± 3.76 (0; 0–12)	0.13 ± 0.35 (0; 0–1)	0.21	1.92 ± 3.27 (1; 0–14)	2.75 ± 3.69 (0; 0–14)	0.44 ± 0.46 (0; 0–1)	0.06	1.58 ± 2.78 (0; 0–8)	2.38 ± 3.09 (0; 0–8)	0.00 (0; 0–0)	0.034
Total active score	0.08 ± 0.28 (0; 0–1)	0.13 ± 0.33 (0; 0–1)	0.00 (0; 0–0)	0.64	0.38 ± 1.01 (0; 0–4)	0.50 ± 1.18 (0; 0–1)	0.13 ± 0.35 (0; 0–1)	0.66	0.25 ± 0.53 (0; 0–2)	0.31 ± 0.59 (0; 0–2)	0.11 ± 0.35 (0; 0–1)	0.53	0.25 ± 0.61 (0; 0–2)	0.38 ± 0.70 (0; 0–2)	0.00 (0; 0–0)	0.30
Total Structural score	0.04 ± 0.20 (0; 0–1)	0.06 ± 0.24 (0; 0–1)	0.00 (0; 0–0)	0.82	1.00 ± 2.59 (0; 0–10)	1.50 ± 3.00 (0; 0–10)	0.00 (0; 0–0)	0.19	1.67 ± 3.10 (1; 0–14)	2.44 ± 3.51 (2; 0–14)	0.33 ± 0.35 (0; 0–2)	0.04	1.33 ± 2.41 (0; 0–8)	2.00 ± 2.69 (1; 0–8)	0.00 (0; 0–0)	0.034
BME	0.04 ± 0.20 (0; 0–1)	0.06 ± 0.24 (0; 0–1)	0.00 (0; 0–0)	0.82	0.38 ± 1.01 (0; 0–4)	0.50 ± 1.18 (0; 0–4)	0.13 ± 0.35 (0; 0–1)	0.70	0.25 ± 0.53 (0; 0–1)	0.31 ± 0.59 (0; 0–2)	0.11 ± 0.35 (0; 0–1)	0.53	0.25 ± 0.61 (0; 0–2)	0.38 ± 0.70 (0; 0–2)	0.00 (0; 0–0)	0.30
Enthesitis	0.04 ± 0.20 (0; 0–1)	0.06 ± 0.24 (0; 0–1)	0.00 (0; 0–0)	0.82	0.00	0.00	0.00	-	0.00	0.00	0.00	-	0.00	0.00	0.00	-
Fat lesion	0.00	0.00	0.00	-	0.29 ± 1.23 (0; 0–1)	0.44 ± 1.46 (0; 0–1)	0.00 (0; 0–0)	0.61	0.96 ± 1.60 (0.0; 0–6)	1.44 ± 1.77 (1; 0–6)	0.22 ± 0.00 (0; 0–2)	0.06	0.38 ± 0.78 (0; 0–3)	0.56 ± 0.80 (0; 0–3)	0.00 (0; 0–0)	0.064
Ankylosis	0.04 ± 0.20 (0; 0–1)	0.06 ± 0.24 (0; 0–1)	0.00 (0; 0–0)	0.82	0.63 ± 2.12 (0; 0–10)	0.94 ± 2.50 (0; 0–10)	0.00 (0; 0–0)	0.44	0.67 ± 2.06 (0.0; 0–10)	0.94 ± 2.42 (0.0; 0–10)	0.11 ± 0.35 (0; 0–1)	0.36	0.83 ± 1.81 (0; 0–6)	1.25 ± 2.07 (0; 0–6)	0.00 (0; 0–0)	0.11
Erosion	0.00	0.00	0.00	-	0.08 ± 0.41 (0; 0–2)	0.13 ± 0.49 (0; 0–2)	0.00 (0; 0–0)	0.81	0.04 ± 0.20 (0; 0–1)	0.06 ± 0.24 (0; 0–1)	0.00 (0; 0–0)	0.81	0.13 ± 0.45 (0; 0–2)	0.19 ± 0.54 (0; 0–2)	0.00 (0; 0–0)	0.61
Sclerosis	0.00	0.00	0.00	-	0.00	0.00	0.00	-	0.00	0.00	0.00	-	0.00	0.00	0.00	-
Capsulitis	0.0	0.0	0.0	-	0.0	0.00	0.00	-	0.0	0.0	0.0	-	0.0	0.0	0.0	-

**Table 4 diagnostics-16-00108-t004:** MRI scores of all patients, and comparison between MeS and non-MeS.

MRI	All (N = 24)	Metabolic Syndrome (N = 15)	No Syndrome (N = 9)	*p* Value
MRI total sum all	5.0 ± 7.2	7.7 ± 8.0 *	0.6 ± 0.8 *	0.013
MRI sum active	0.96 ± 1.52	1.40 ± 1.76	0.22 ± 0.44	0.08
MRI sum structural	4.04 ± 6.39 *	6.27 ± 7.26 *	0.33 ± 0.71	0.014
Bone marrow edema (median; range)	0.92 ± 1.53	1.33 ± 1.80	0.22 ± 0.44	0.14
Enthesis (median; range)	0.04 ± 0.20	0.07 ± 0.26	0.00 ± 0.00	0.82
Fat lesion (median; range)	1.63 ± 2.72	2.47 ± 3.13 *	0.22 ± 0.67 *	0.013
Ankylosis (median; range)	2.17 ± 4.78	3.40 ± 5.77	0.11 ± 0.33	0.1
Erosions (median; range)	0.25 ± 0.68	0.40 ± 0.82	0.00 ± 0.00	0.31
Sclerosis	0.0	0.0	0.0	-
Capsulitis	0.0	0.0	0.0	-

* Statistically significant for *p* Value < 0.05. Association between MRI scores and clinical parameters.

## Data Availability

The raw data is not accessible to the public yet can be accessed with permission from the authors and Clalit Health Services data access committee.

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
