# Peer review of "Whole Spine and Sacroiliac Joint MRI of Patients With and Without Metabolic Syndrome: A Preliminary Study"

_diagnostics, 2025, doi:10.3390/diagnostics16010108_

Round 1
Reviewer 1 Report
Comments and Suggestions for Authors
This is an interesting, novel, and potentially impactful preliminary study exploring whether metabolic syndrome (MeS) is associated with early MRI-detectable spinal and SIJ lesions suggestive of a “pre-DISH” phenotype. The topic is timely given rising global MeS prevalence and increasing interest in DISH pathogenesis. The study is clearly written, clinically relevant, and uses standardized MRI scoring.
- The MeS (n=15) and non-MeS (n=9) groups differ strongly. These profound metabolic differences make it impossible to attribute MRI differences. BMI alone is known to cause fatty spinal marrow and structural lesions.
- Consider performing adjusted analyses (ANCOVA or regression) controlling for BMI and waist circumference.
- Authors used one-sided p-values stating “direction of effect predetermined.”
This is not acceptable in exploratory imaging research with no prior quantitative evidence. - Given the number of MRI variables (dozens of regions × many lesion types), the study is severely underpowered, increasing type I and type II error.
- No longitudinal imaging, No biochemical markers of bone turnover, No correlation with ossification progression for suggest pre-DISH statte
- Inclusion criteria allowed CT/radiograph up to 12 months prior to MRI but DISH progression over one year is documented.
Author Response
We thank the reviewers for their commitment to improve our manuscript in order to be suitable for publication. We thank the reviewers for their work and will answer their comments in a point by point answers.
Our answers will be in italic under-line text
The changed text will be presented here in bold format
Reviewer 1:
This is an interesting, novel, and potentially impactful preliminary study exploring whether metabolic syndrome (MeS) is associated with early MRI-detectable spinal and SIJ lesions suggestive of a “pre-DISH” phenotype. The topic is timely given rising global MeS prevalence and increasing interest in DISH pathogenesis. The study is clearly written, clinically relevant, and uses standardized MRI scoring.
We thank the reviewer for this opinion.
Reviewer 1 comment 1:
The MeS (n=15) and non-MeS (n=9) groups differ strongly. These profound metabolic differences make it impossible to attribute MRI differences. BMI alone is known to cause fatty spinal marrow and structural lesions.
Answer : We thank the reviewer for this point. We added this as a limitation to the text
Because our groups were small and we analyzed multiple MRI lesion types across several regions, the MRI changes reported here may not be directly attributable to MeS itself. Instead, they could reflect the influence of individual MeS components and the differences in these components between groups. Accordingly, these findings are susceptible to both type I and type II errors and should be interpreted with caution.
Page 12 lines 406-410
Reviewer 1 comment 2:
Consider performing adjusted analyses (ANCOVA or regression) controlling for BMI and waist circumference.
Answer : We thank the reviewer for this point, Yet, we think that correcting for waist circumference and/or BMI is incorrect as obesity is a component of metabolic syndrome, which is the effector of this study’s hypothesis.
Reviewer 1 comment 3:
Authors used one-sided p-values stating “direction of effect predetermined.”
This is not acceptable in exploratory imaging research with no prior quantitative evidence.
Answer : We thank the reviewer for this point. We have re-checked this with our statistician and the reported results and with 2-sided p-values. Another version we used a one sided and accidentally did not change it in the manuscrpipt. The data for one sided p value was omitted and carful overview of all the presented results here was re done. Text was changes accordingly.
Two-sided analysis was used as the direction of effect was not pre-determined.
Page 4 line 212
Reviewer 1 comment 4:
Given the number of MRI variables (dozens of regions × many lesion types), the study is severely underpowered, increasing type I and type II error.
Answer : We thank the reviewer for this point. We added this as a limitation, please refer to comment 1 as it was changed accordingly to that as well.
Reviewer 1 comment 5:
No longitudinal imaging, No biochemical markers of bone turnover, No correlation with ossification progression for suggest pre-DISH statte
Answer : We thank the reviewer for this point. As this is the data we have collected, and since our study is not a longitudinal one, we added this important points to the limitations. We hope to do a follow-up study in order to address these issues.
Finally, as this was a cross-sectional analysis at one point of time, no longitudinal was available to address the important issue of ossification progression and its biomarkers.
Page 13 line 429-431
Reviewer 1 comment 6:
Inclusion criteria allowed CT/radiograph up to 12 months prior to MRI but DISH progression over one year is documented.
Answer : We thank the reviewer for this point.Yet, the aim was not to search for DISH in a previous pre-DISH XR or CT. the 12 months timeframe was to assure that these patients are at no-DISH or pre-DISH state, as we hypothised that MeS is the driving force for ossification. We added this clarification in text:
Also, as progression to DISH takes years, this study could not address that since we al-lowed no more than 12 months of CT or XR before inclusion.
Page 13 line 431-433
Reviewer 2 Report
Comments and Suggestions for Authors
The manuscript addresses an interesting topic; however, it presents substantial methodological limitations:
- The musculoskeletal complaints are not clearly specified, and spondyloarthritis has not been fully ruled out. In addition, no correlation of clinical symptoms with MRI findings has been established.
- Statistical analysis: Given that the sample size is below 30 patients, the authors should use Fisher’s exact test or Yates’ correction. The chi-square test is not appropriate in this context.
- A comparison group of patients with spondyloarthritis should be included and evaluated using the same MRI technique and scoring system to allow for an objective assessment and comparison.
- The tables need to be better formatted and presented more clearly.
Best regards.
Author Response
We thank the reviewers for their commitment to improve our manuscript in order to be suitable for publication. We thank the reviewers for their work and will answer their comments in a point by point answers.
Our answers will be in italic under-line text
The changed text will be presented here in bold format
Reviewer 2 comment 1:
The musculoskeletal complaints are not clearly specified, and spondyloarthritis has not been fully ruled out. In addition, no correlation of clinical symptoms with MRI findings has been established.
Answer: We thank the reviewer for this point. MSK complaints were screened at inclusion only to exclude inflammatory back pain, This study was not assessing clinical symptoms hence there are no correlations. Clarifications were made in the text.
Another limitation is that participants were recruited through a rheumatic disease unit. Although enrollment was voluntary and included individuals both with and without back pain, this recruitment strategy may have introduced selection bias. To reduce this risk, we systematically evaluated for inflammatory back pain and excluded participants who fulfilled established criteria.
Page 13 line 418 – page 14 line 429
Reviewer 2 comment 2:
Statistical analysis: Given that the sample size is below 30 patients, the authors should use Fisher’s exact test or Yates’ correction. The chi-square test is not appropriate in this context.
Answer: We thank the reviewer for this point. We did re-analysis using Fisher’s exact test, Changes are implemented in text and tables. No major changes from previous analysis were found.
…Demographic and clinical characteristics of the two groups were com-pared by Fisher’s exact test for….
Page 4 line 208
Reviewer 2 comment 3:
A comparison group of patients with spondyloarthritis should be included and evaluated using the same MRI technique and scoring system to allow for an objective assessment and comparison.
Answer: We thank the reviewer for this point. As this is a preliminary relatively small study which involves whole spine MRI voluntarily done by the patients, we could not add another cohort of AxSPA patients since it will demand changes in IRB and another session of recruitment, We thank the reviewer for the idea which we will try to incorporate in following works of our group.
Reviewer 2 comment 4:
The tables need to be better formatted and presented more clearly.
Answer: We thank the reviewer for this point, Changes were made in table 3, and figures were added.
Reviewer 3 Report
Comments and Suggestions for Authors
Introduction
- Conflates inflammatory and structural lesions without clearly distinguishing their relevance to MeS vs. DISH pathophysiology.
- Review of MRI findings in established DISH is accurate, but prior cohort descriptions could be synthesized rather than detailed individually.
Materials and Methods
- The study description “descriptive work comparing frequency of MRI lesions” is vague; clarify whether the design is cross-sectional, observational, or case–control.
- Inclusion/exclusion criteria partially clear; inconsistencies remain:
- Age range narrow (40–50 years), atypical for DISH/MeS cohorts; requires justification.
- Recruitment mixes symptomatic and asymptomatic volunteers, introducing potential selection bias; should be acknowledged.
- Clinical variables collected are appropriate, but timing relative to MRI (e.g., same visit, fasting) is unclear.
- The manuscript cites the Berlin scoring method; however, the described scoring system (combined SIJ/spine assessment, binary lesion scoring, point allocation) differs substantially from the validated Berlin MRI score (ASAS/OMERACT).
- It appears to be an ad-hoc scoring system developed for this study rather than direct application of the Berlin method.
- Recommend clarification of the rationale behind this modified score and its conceptual/methodological relation to the original Berlin score.
- Include example MRI images from both groups with explanations of the scoring; in the article there are no MRI images, even if it is a radiology article.
Results
- Paragraph is overly concise and relies heavily on tables.
- Tables are poorly formatted; large tables should be displayed in landscape format on full pages if necessary.
Discussion
- References to DISH studies are appropriate; however, comparison between a “pre-DISH” cohort and established DISH populations is inherently limited.
- Lacks discussion of studies correlating MRI findings with systemic adipoinflammation, which would strengthen biological plausibility. If not available, this limitation should be explicitly stated.
Author Response
We thank the reviewers for their commitment to improve our manuscript in order to be suitable for publication. We thank the reviewers for their work and will answer their comments in a point by point answers.
Our answers will be in italic under-line text
The changed text will be in bold
Reviewer 3 comment 1:
Introduction - Conflates inflammatory and structural lesions without clearly distinguishing their relevance to MeS vs. DISH pathophysiology. Review of MRI findings in established DISH is accurate, but prior cohort descriptions could be synthesized rather than detailed individually.
Answer : We thank the reviewer for this point. We have changed the introduction accordingly.
Both DISH and axial spondyloarthritis (axSpA) are enthesis-related, bone-forming conditions that in their end-stage may lead to spinal ankylosis. [11] Nevertheless, their pathophysiology is thought to differ substantially: DISH is dominated by structural new bone formation, whereas axSpA is characterized by inflammation-driven lesions that may precede structural change. [12] MRI lesions can therefore be conceptually grouped into inflammatory (active) findings—such as bone marrow edema (BME), enthesitis and capsulitis—and structural findings—such as fat lesions, sclerosis, erosions and ankylosis.
In established DISH, prior MRI studies have mainly described structural corner-based changes (particularly fat lesions and sclerosis), with variable presence of BME, and overall patterns that can differ from axSpA/AS particularly in the distribution and extent of ankylosis. [13] However, these reports address established DISH, and do not clarify whether similar MRI abnormalities can be detected earlier in individuals with MeS in the absence of radiographic/CT evidence of DISH.
Spinal MRI of patients with MeS has been explored in limited settings, including postmenopausal women with DM, where lumbar MRI spectroscopy suggested altered marrow lipid composition and higher marrow density consistent with the skeletal effects of DM. [14,15] In addition, ultrasound studies suggest that MeS may be associated with a pro-inflammatory entheseal phenotype, and a proportion of these patients also fulfill DISH criteria. [16]
The presence and pattern of spinal MRI lesions in patients with MeS without im-aging evidence of DISH (i.e., “pre-DISH” candidates) has not yet been studied. This study aims to describe patterns of MRI lesions among relatively young MeS patients with no imaging evidence of DISH. We hypothesized that the MeS group will have higher MRI lesion scores compared to the non-MeS group
Page 2 line 44-67
Reviewer 3 comment 2:
Materials and MethodsThe study description “descriptive work comparing frequency of MRI lesions” is vague; clarify whether the design is cross-sectional, observational, or case–control.
Answer : We thank the reviewer for this point. We have changed it accordingly.
This is a cross-sectional study, comparing the frequency of MRI…
Page 3 line 137
Reviewer 3 comment 3:
Inclusion/exclusion criteria partially clear; inconsistencies remain: Age range narrow (40–50 years), atypical for DISH/MeS cohorts; requires justification. Recruitment mixes symptomatic and asymptomatic volunteers, introducing potential selection bias; should be acknowledged.
Answer : We thank the reviewer for this point, we addressed those issues and corrected in text.
We focused on this relatively narrow age range because DISH is presumed to be less common at these ages, whereas MeS is still prevalent, thus allowing MeS to be evaluated as a potential pre-DISH state.
Page 3 line 147-149
And
Another limitation is that participants were recruited through a rheumatic disease unit. Although enrollment was voluntary and included individuals both with and without back pain, this recruitment strategy may have introduced selection bias. To reduce this risk, we systematically evaluated for inflammatory back pain and excluded participants who fulfilled established criteria.
Page 12 line 418- page 13 line 429
Reviewer 3 comment 4:
Clinical variables collected are appropriate, but timing relative to MRI (e.g., same visit, fasting) is unclear.
Answer : We thank the reviewer for this point, we addressed those issues and clarified in text.
All clinical data was collected from electronic medical records with data available in a time frame of 6 months before and after the study MRI.
Page 4 line 204
Reviewer 3 comment 5:
The manuscript cites the Berlin scoring method; however, the described scoring system (combined SIJ/spine assessment, binary lesion scoring, point allocation) differs substantially from the validated Berlin MRI score (ASAS/OMERACT).
It appears to be an ad-hoc scoring system developed for this study rather than direct application of the Berlin method.
Recommend clarification of the rationale behind this modified score and its conceptual/methodological relation to the original Berlin score.
Answer : We thank the reviewer for this point, we addressed those issues and corrected in text.
…Because the Berlin scoring system was originally developed to quantify spinal and sacroiliac joint inflammation in axial spondyloarthritis, its grading relies on the presence of relatively large and frequent inflammatory lesions. In contrast, patients with DISH typically demonstrate much smaller, more focal, and less frequent lesions, making the full 0–3 Berlin scale disproportionate for this population. To maintain the same anatomical topography and standardized assessment framework while ensuring sensitivity to the subtle lesions characteristic of DISH, we adapted the scoring as follows: sacroiliac joints were scored on a binary 0–1 scale (absent/present), and vertebral corner lesions in the spine on a 0–2 scale reflecting absent, mild, or definite focal abnormalities. This modification preserves consistency with established SpA scoring methodology while improving its applicability and discriminatory capacity in DISH.
Page 3 line 164-175
Reviewer 3 comment 6:
Include example MRI images from both groups with explanations of the scoring; in the article there are no MRI images, even if it is a radiology article.
Answer : We thank the reviewer for this point, we added three figures with detailed legends.
Reviewer 3 comment 6:
Results
- Paragraph is overly concise and relies heavily on tables.
- Tables are poorly formatted; large tables should be displayed in landscape format on full pages if necessary.
Answer : We thank the reviewer for this point, We detailed the text of results, and reshaped the tables to a better format. Here is the revised first paragraph, We assume that the reviewer addressed this paragraph specifically since other results paragraph regarding imaging were detailed.
Twenty-four patients were included (13 males; mean 45.5 years at imaging, range 40-49 years), 15 in the MeS group and 9 in the non-MeS group. The cohorts’ demographics, medical history, medications and blood chemistry data are presented in Table 1 and Table 2. The mean age was 44.9 ± 3.1, without statistically significant difference. Among MeS group 9 out of 15 were males compared to controls with 3 out of 9, without statistically significant difference. Patients with MeS had a statistically significantly higher rate of all MeS components (i.e higher triglycerides and lower HDL, higher frequency of DM and IFG, larger waist circumference, and significantly higher BMI (34.9 Vs. 22.7). There were no other statistically significant differences in medical history or any significant differences in medication uptake between the two groups. Patients with MeS also had significantly higher median insulin, Uric Acid, AST, ALT, ESR and CRP levels
Page 4 line 225-229
Discussion
Reviewer 3 comment 7:
References to DISH studies are appropriate; however, comparison between a “pre-DISH” cohort and established DISH populations is inherently limited.
Answer : We thank the reviewer for this point, we added our comments on that in text.
It should be noted that that not all MeS are also defined as pre-DISH state, as well as not all pre-DISH will eventually progress to DISH respecting Resnick’s criteria. Hence, the results of this study focus mostly on the association between MeS and spinal MRI changes which might precede pre-DISH and DISH.
Page 11 line 336-340
Reviewer 3 comment 8:
Lacks discussion of studies correlating MRI findings with systemic adipoinflammation, which would strengthen biological plausibility. If not available, this limitation should be explicitly stated.
Answer : We thank the reviewer for this point, we added our comments on that in text.
Evidence from systematic evaluations of spinal and SIJ MRI changes in patients with MeS remains limited. Badr et al. used MRI with spectroscopy to assess bone marrow adiposity in postmenopausal women with type 2 DM, demonstrating lower levels of unsaturated fat in the lumbar spine in this population. In addition, Woods et al. reported a negative correlation between unsaturated fat levels in bone marrow and fracture risk. However, neither study assessed concomitant morphological spinal changes or soft-tissue alterations related to adiposity. Moreover, no additional studies evaluating the association between MeS and MRI-detected spinal changes were identified, positioning our work as preliminary and underscoring the need for confirmation in larger, well-characterized cohorts.
Page 12 line 392-401
Round 2
Reviewer 1 Report
Comments and Suggestions for Authors
The manuscript can be published in its current form.